# Lawn Lamp Design Based on Fuzzy Control and Secondary Optical Optimization

Xinjing Qin, Zhisheng Wang *, Manqun Zhang, Yue Feng and Kexian Li

Research Institute of Photonics, Dalian Polytechnic University, Dalian 116034, China
* Correspondence: wangzs@dlpu.edu.cn

**Abstract:** With the emergence of new technologies, the design of urban infrastructure is constantly being innovated, and the lawn lamp as urban lighting infrastructure is an important part of urban infrastructure. For the current lawn lamp function, there are single, large power consumption, low light energy utilization and other shortcomings. Combined with deep learning and optical design, this paper constructs an adaptive lighting control system based on the technology of the Internet. Considering the nonlinear and time-varying characteristics of external factors, a fuzzy control model with ambient light level and pedestrian flow as input and dimming coefficient $K$ ($0 < K < 1$) as output is proposed to adjust the brightness of the light source and achieve energy savings. In order to improve the light energy utilization of the luminaire and reduce the glare index of the luminaire, a free-form total internal reflection (TIR) lens was designed by finding the optimal curvature of the lens through the polycurved edge light principle. The light source of the lawn lamp was simulated by TracePro, and the results showed that the light energy utilization reached 90%. Finally, the ambient illumination and pedestrian flow data of Dalian ZT Park were measured for different time periods at the site, and the data were normalized using the min-max normalization algorithm. The adaptive dimming capability of the system was verified through simulation tests and field tests, and the results showed that the lighting energy efficiency under the control system was 38%.

**Keywords:** smart systems; fuzzy algorithm; TIR lens; dimming coefficient

## 1. Introduction

Outdoor lighting is one of the main energy-consuming products in the lighting industry, accounting for about 40–50% of the total consumption [1]. Lawn lamps, as a kind of outdoor lighting equipment, have been widely used, and in Europe, the lawn coverage rate is high. Coupled with the European promotion of energy savings and environmental protection, the installation of solar lawn lamps in the green landscape has become a trend [2]. As the level of urban greening continues to increase, the demand for outdoor lawn lamps is growing, and there are a variety of lighting products on the market [3]. Energy waste is ubiquitous in these lighting systems because all lawn lights are lit regardless of the presence of pedestrians or changes in daylight, which can lead to serious over-illumination problems. Therefore, it is of great importance to design an energy-efficient outdoor lighting control system under the premise of ensuring traffic safety. For outdoor lighting, scholars have mostly focused on street lighting as well as tunnel lighting, with little research on decorative luminaires. Tan et al. fully integrated the information of weather, sunrise and sunset, and road traffic flow at night in the design process to achieve energy savings by means of double fuzzy controller switching. It provides a new way of thinking and method for energy savings in urban lighting systems. It makes up for the deficiencies of the traditional single fuzzy controller method [4]. Wang et al. designed an intelligent LED lighting system based on the STC89C52 microcomputer to achieve light intensity control and infrared detection but ignored the analysis of experimental results [5]. Lu et al. used solar LED lighting technology for basic and emergency lighting of luminaires. However,

solar tracking devices and fiber optic lighting systems are relatively expensive and not applicable to lawn lights [6]. Du et al. designed a control strategy for lighting with fuzzy (PID) control as the dimming method to solve the problem of high power consumption in tunnel lighting. The experimental results show that the system can effectively reduce the impact of the light decay phenomenon on the lighting control system to ensure the safety of traffic [7]. Yang et al. proposed a fuzzy control method for tunnel lighting control systems. Simulation results show that the fuzzy system has a significant energy saving effect and good adaptability [8]. In recent years, on the basis of energy savings, outdoor lighting has been more inclined to meet the comfort needs of users [9,10].

As a kind of urban infrastructure serving human society, lawn lamps will be directly related to the safety and quality of people's lives. The operation of new technologies and the design thinking of new media can make lawn lights have a good interaction with users. Traditional lawn lights are mainly used to achieve basic functional lighting, such as illuminating roads or lawns, and they lack interaction with people, so an optimal redesign of them is necessary. It is important to note that as an outdoor luminaire, the loss of lamps is very high, and the lighting effect of lamps decreases over time. Therefore, this study breaks the original lawn lamp design concept and adopts an optical design to improve the lighting effect of the lawn lamp and reserve more space for dimming for intelligent control. In this paper, we use a fuzzy control algorithm, taking ambient light level and pedestrian flow as inputs and the dimming coefficient $K$ ($0 < K < 1$) as output, to solve the problem of constant light from traditional lamps. In order to improve the light energy utilization of lamps and lanterns and reduce the number of glare values of lamps, this study carries out secondary optical design of light sources and designs a collimation system based on a total internal reflection (TIR) lens according to the principle of edge light and synchronous polysurface design method, which controls the light output direction by the TIR lens and collimates the light to make the light source achieve the expected effect.

## 2. Overall Design

### 2.1. Research Background

In this study, ZT Park in Changhai County, Dalian, China, was selected as the research object. The sidewalk of the park is 1038 m long and 1.5 m wide, and 173 lawn lamps are required to be installed, assuming the installation interval of lawn lights is 6 m. The lawn lamps are arranged on both sides of the road in a double-sided cross arrangement with the road centerline as a reference, and the lawn lamps on both sides are distributed at different points corresponding to the centerline. This arrangement has the advantage of good horizontal uniformity of lighting brightness. The layout of the scenes and lamps is shown in Figure 1.

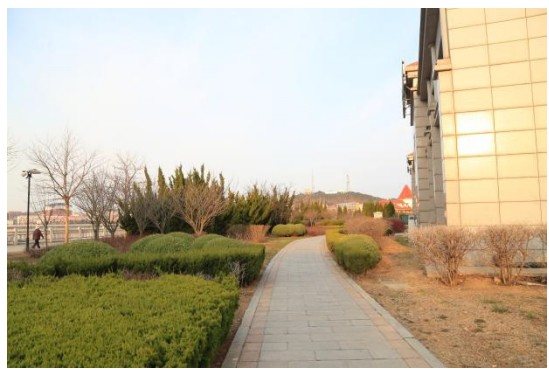 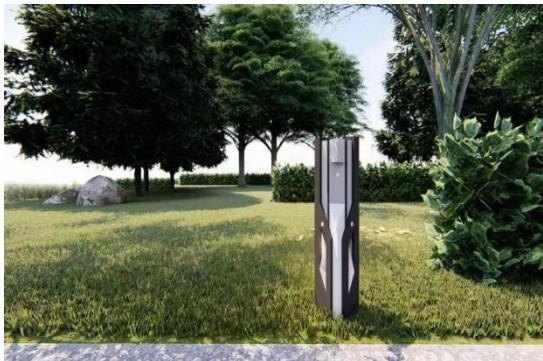

**Figure 1.** The diagram of park sidewalk scene and lighting layout effects.

In addition, according to the municipal road regulations, when the period is 6:00 p.m.–9:00 p.m., the park has a high flow of people, and all the outdoor lamps are turned on. When the night is 9:00–12:00, taking into account the energy-saving lighting and

low traffic, ornamental lamps are closed, leaving only peripheral street lights and sidewalk lawn lamps to ensure the safety of pedestrians traveling at night [11].

Any lighting design needs to be in accordance with standards, but there are no clear standard regulations for lawn lamps. The lighting design of the lawn lamp is related to the adjacent street lights, which are half of the adjacent non-motorized road illumination and not less than 5 lx. Therefore, the lower limit of the dimming coefficient should be accurately set during the lighting design.

### 2.2. Control System Design

Traditional lawn lamps are generally controlled by timing, which consumes more energy [12]. The existing lawn lamps only satisfy the lighting function, which does not reflect the concept of a smart city. With the increase in outdoor lighting types, natural illumination is no longer the only factor affecting lawn dimming but also needs to consider the ambient light level brought by the surrounding lamps. Therefore, in this study, in order to optimize the above problems, an intelligent lawn lamp with adaptive dimming that combines ambient light level and pedestrian flow is studied. The lawn lamp can collect the required data through a light sensor and pedestrian flow sensor, and after the control system analyzes and processes the best dimming coefficient of the lamps and lanterns to realize multi-light control through centralized control. The control system can be divided into three layers: the data acquisition layer, the data transmission layer, the data processing and the display layer. The architecture of the control system is shown in Figure 2.

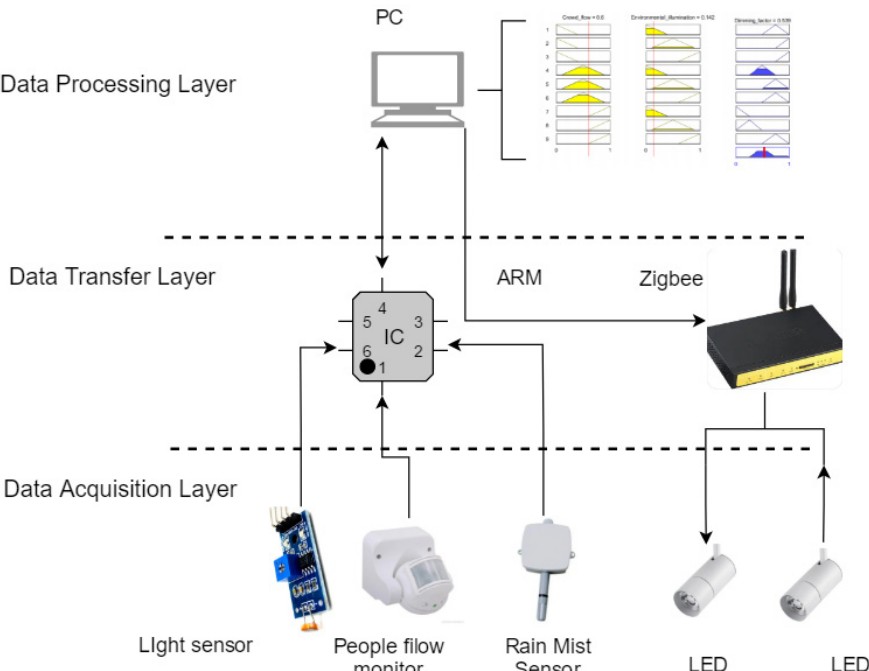

**Figure 2.** The architecture of the control system.

(1)　Data acquisition layer: this layer collects data such as ambient light level, pedestrian flow, weather factors and electronic ballasts for controlling LED luminaires.
(2)　Data transmission layer: This layer consists of an ARM embedded system and a Zigbee wireless transmission module. The ARM embedded system collects the data collected by the sensors for transmission to the fuzzy controller.
(3)　Data processing and display layer: This layer includes the fuzzy controller that uses the given fuzzy logic to calculate the output dimming coefficients.

In this paper, we also study the effect of CCT (color temperature) on human emotions, as good lighting design aims to create a light environment that is beneficial to human emotions and can sometimes even improve work efficiency and increase happiness [13].

The light source is low color warm light, which makes the interior space look warm, relaxed and stable, while high CCT and cool light will create a fresh and energetic atmosphere in the space. The park has a high flow of people from 5:00 p.m. to 10:00 p.m., which is a relaxing space for the busy workers, so the CCT is 3500 K to create a warm light environment. When the time is 4:00 a.m., it is suitable to adjust the CCT higher, so that people produce a fresh, clean feeling. On the other hand, coastal areas have more rain and fog, which will directly affect the lighting effect of the lawn lamp due to the low CCT of the light and its strong penetrating power, so in the rain and fog, control light source output with a low CCT, the process of CCT adjustment as shown in Algorithm 1.

---

**Algorithm 1** The Process of CCT Adjustment

---

**Input:**
    Time period, weather factors;
**Output:**
   Change of CCT;
    1: Initialize hardware devices;
    2: Acquisition of data;
    3: **for** Number of time judgments **do**
    4:      Time judgment;
    5:      if (17:00 < =Time < =22:00)||(Time >= 04:00)
    6:        **for** Number of weather factors judgments **do**
    7:          Weather judgment;
    8:          if(rain) ||(fog)
    9:        end
  10: end

---

The lighting control method in this study is in the form of one-to-many, i.e., the centralized controller is externally connected to facilities, such as environmental information collection sensors and pedestrian flow collection sensors, and the sensors will upload the real-time collected data to the centralized controller to achieve the dimming coefficients of the lamps through the fuzzy controller. The physical location of the centralized controller is generally arranged in front of its management area, and the thesis will investigate the lighting control strategy for this one-to-many form, which can be used for the corresponding brightness adjustment control operation of multiple lamps.

### 2.3. Fuzzy Control

Since the lawn lamp dimming system in this study is determined by the ambient light level and the number of pedestrians, these random, unpredictable factors have nonlinear and time-varying characteristics on the dimming of the lamps [14]. Therefore, the judgment made by the traditional controller only by analyzing the sensor data has errors and cannot adapt to this nonlinear, multivariable natural system. In contrast, fuzzy control can translate human control experience, expertise and reasoning decisions into automatic control methods that are suitable for variable optimization. In order to achieve the most suitable illumination level for pedestrians and save energy, this paper uses fuzzy control to regulate the luminous intensity of the lawn lamp. The fuzzy control principle is shown in Figure 3.

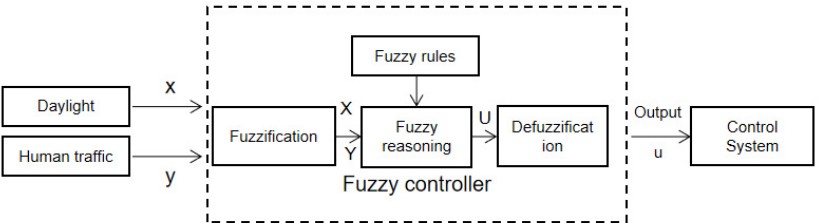

**Figure 3.** Fuzzy control model diagram.

As shown in Figure 3, the fuzzy control is divided into three parts, including the input, the fuzzy control algorithm and the output. In this study, ambient illumination and pedestrian flow are used as the input of fuzzy control and are represented by x, y and the dimming coefficient K of the light source is represented by u as the output. The input variables x and y are fuzzified (D/F) to obtain the fuzzy quantities X and Y.

### 2.3.1. Fuzzification

The fuzzy control quantity U is obtained by reasoning based on empirical fuzzy control rules, and then U is defuzzified according to a certain algorithm and converted into a dimming coefficient K(u) that can be used for dimming. The basic domain of the input and output is the following equation.

$$\begin{cases} x \in [0, 100]\,(\%) \\ y \in [0, 100]\,(\%) \\ u \in [0.2, 1] \end{cases} \tag{1}$$

The actual input and output values are converted to fuzzy input and output by scale transformation. Suppose $X_{\text{physical}}$ is the actual input or output value, $X_{\text{fuzzy}}$ is the fuzzy input or output value, and the relationship between the two can be obtained by the extreme value method.

$$X_{\text{fuzzy}} = \frac{X_{\text{min}} + X_{\text{max}}}{2} + \frac{X_{\text{max}} - x_{\text{min}}}{X^*_{\text{max}} - x^*_{\text{min}}}(X_{\text{physcial}} - \frac{X^*_{\text{main}} + X^*_{\text{max}}}{2}) \tag{2}$$

where, $X_{\text{fuzzy}} \in [X_{\text{min}}, X_{\text{max}}]$, $X_{\text{physical}} \in [X^*_{\text{min}}, X^*_{\text{max}}]$.

For the fuzzy input $X$ of natural light, three levels of SD (dark), MD (moderate) and LD (bright) are set. Similarly, for the fuzzy input Y of pedestrian flow, three levels are set: SG (little), MG (medium) and LG (much). The output light source brightness is divided into five fuzzy sets: VS (very small), S (small), M (medium), L (large) and VL (very large). Fuzzy control rules are the core of fuzzy controller design, which consists of fuzzy conditional control statements summarized by experience or operators such as the structure of "if A and B then C" to complete the entire design of fuzzy rules [15]. For example, the first rule is that if the ambient illumination is dark and the pedestrian flow is low, the output dimming coefficient K is large. The fuzzy rule table for this study is shown in Table 1.

**Table 1.** Fuzzy rule distribution.

| Luminous Flux (L) | | Crowd Flow | | |
|---|---|---|---|---|
| | | SG | MG | LG |
| | SD | S | L | VL |
| Natural Light | MD | S | M | M |
| | LD | VS | S | S |

### 2.3.2. Fuzzy Inference

The algorithm of fuzzy inference is directly related to the setting of fuzzy rules, and its complexity depends on the determination of the affiliation function of the fuzzy set in

the fuzzy rule statement. This module is designed using the commonly used Mamdani inference algorithm, and the Matlab fuzzy tool is used to configure the affiliation function and inference rules to obtain the relational surface of the input and output, as shown in Figure 4.

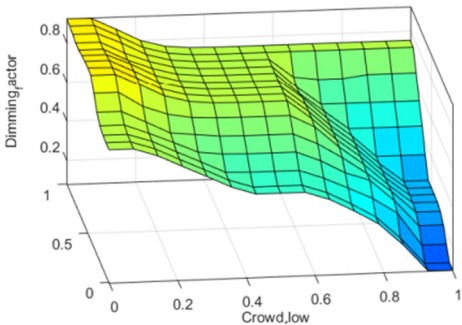

**Figure 4.** 3-D visualization of the inference rules.

### 2.3.3. Defuzzification

The result of fuzzy inference is a fuzzy value, which needs to be non-fuzzy to achieve the deterministic output value to be used for the corresponding control of the luminaire, and this paper uses the region-center method to convert the fuzzy output into the actual dimming coefficient. The formula is shown as follows:

$$U = \frac{\sum_1^i \mu(u_i)u_i}{\sum_1^i \mu(u_i)} \tag{3}$$

where $\mu(u_i)$ is the affiliation function and $u_i$ is the center of the Max-Min combination of the output affiliation function. $u$ is the comfortable illumination value.

### 3. Secondary Optical Design

LED has the advantages of a long life, safety in production and use, environmental protection and high luminous efficiency, so the lawn lamp in this study uses LED as the light source [16]. On the one hand, LEDs have a Lambertian luminescence, with a large light angle, strong light in the middle and small light around them and uneven light output. On the other hand, the lawn lamp in this study also needs to consider the light source to avoid bringing glare to pedestrians and improve the light energy utilization. Therefore, the secondary optical design of the LED light source is needed. The secondary optical design uses the polysurface calculation method to find out the discrete points of the surface and then after fitting to form the refractive surface. Using the optical simulation Tracepro software (TracePro70 (CHS), Lambda Research Corporation, Littleton, MA, USA), the light from the LED is redistributed by designing the reflector and lens to adjust the light output effect.

### 3.1. TIR Lens Design

A total internal reflection (TIR) lens is a special free-form lens whose main function is designed to collect the light emitted from the LED light source as much as possible, while at the same time reducing the divergence angle of the light beam and reducing the numerical aperture, in order to achieve high brightness illumination of the lighting device. Facilitating the design of subsequent optical systems and improving the light energy utilization rate are also improved by reducing the numerical aperture to achieve high-brightness illumination of the illumination device and the subsequent optical system design [17]. Since free-form TIR lenses obtain the surface shape of the lens by tracing the travel path of each light ray, it is difficult to obtain the structure of the lens as a function of the illumination requirements. Therefore, the initial structure of the lens surface is obtained by the theoretical calculation of each light path. The working principle is shown in Figure 5.

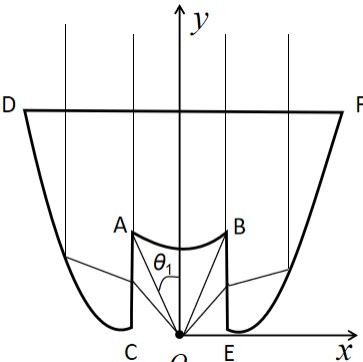

**Figure 5.** Optical design working principle diagram.

Since the LED has rotational symmetry, the lens surface can be composed of a 360° rotation of the cross section around the *y*-axis, and the LED light source is represented by the *O* point. When the angle of incidence $\theta$ ($0 \leq \theta \leq \theta_1$), the light of the LED is only transmitted by the surface *AB* to become parallel light. When the angle of incidence $\theta$ ($\theta_1 \leq \theta \leq 90°$), the LED light is refracted by the rotating surfaces *AC* and *BE* and then reflected by the rotating surfaces *CD* and *EF* to become parallel light. Therefore, the TIR lens is designed in two aspects according to the magnitude of the incidence angle.

### 3.1.1. Intermediate Aspheric Refractive Surface Calculation

Since the lenses are symmetrical to each other, the right half will be used for the design, and the design schematic is shown in Figure 6.

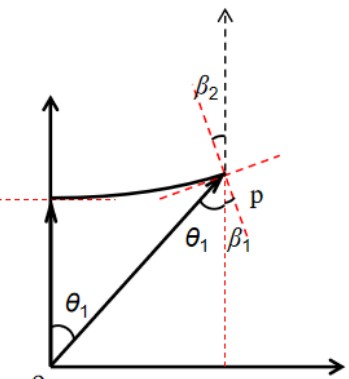

**Figure 6.** Central non-sheik surface optical design.

First, the starting point $P(x_i, y_i)$ is determined on the central aspherical plane, which is the highest point of the refractive surface of the lens. The angle of incidence is ($\theta_1 + \beta_1$) and the angle of emergence is $\beta_2$. Assuming that the incident medium is air, whose refractive index is expressed by $n_0$, and the emergent medium is the lens material $n_1$, according to the law of refraction, it is obtained that,

$$n_0 \sin(\beta_1 + \theta_1) = n_1 \sin \beta_2 \tag{4}$$

The curve is an accumulation of countless points, so the coordinates of the next point are derived from the initial coordinates. According to the trigonometric transformation:

$$\frac{dy_i}{dx_i} = \frac{y_i - y_{i-1}}{x_i - x_{i-1}} = \tan(90 - \theta_1) \tag{5}$$

where $(x_{i-1}, y_{i-1})$ are the coordinates of the next point. Combining the above two formulas, the iterative relationship between the coordinates of a point and the initial coordinates and the angle of incidence is shown in the following equation.

$$y_{i-1} = \frac{y_i - k_i x_i}{1 - \tan \theta_{i-1} k_i} \tag{6}$$

$$x_{i-1} = \frac{\tan \theta_{i-1} (yi - k_i x_i)}{1 - \tan \theta_{i-1} k_i}$$

The calculation method proposed in this design can be studied according to the demanded size of the lens. Firstly, the size of the intermediate non-surface is given, and the horizontal coordinate $x_i$ is assumed. The vertical coordinate is obtained according to the trigonometric transformation, and the point is the critical point between the intermediate non-spherical surface and the edge total reflective surface. Finally, according to the above formula, the discrete point of this surface can be found by constant iteration.

### 3.1.2. Design of Side Total Reflective Surfaces

When the light source divergence angle ($\theta_1 \leq \theta \leq 90°$), the rotating surface *AB* does not play an optical role, and the light rays become parallel rays by refracting through the rotating surfaces *AC* and *BE* and then reflecting through the rotating surfaces *CD* and *EF*. Therefore, the curvature of the outer surface needs to be calculated. The design of the lateral total reflective surface is shown in Figure 7, and a point *P* is taken on the curve *EF*, according to the law of refraction:

$$n_0 \sin (90 - \theta) = n_1 \sin (\beta') \tag{7}$$

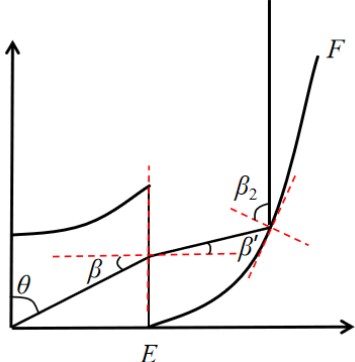

**Figure 7.** Side reflective surface optical design principle.

According to the sum of the interior angles, the triangle is 180°, the angle of reflection: $\beta_2 = \frac{\beta' + 90°}{2}$

Suppose the horizontal coordinate of point *P* is $x_i$, then the vertical coordinate $y_i = x_i \tan(90° - \theta)$, from the definition of slope,

$$\frac{y_{i+1} - x_0 \tan(90 - \theta)}{x_{i+1} - x_i} = \tan \beta' \tag{8}$$

According to the iterative calculation,

$$x_{i+1} = \frac{x_0 \tan \beta'_{i+1} - k_i x_i - x_0 \tan(90 - \theta_{i+1})}{\tan \beta'_{i+1} - k_i} \tag{9}$$

$$y_{i+1} = \frac{K_i x_0 \tan(90 - \theta_{i+1}) - yi \tan \beta'_{i+1} + k_i x_i \tan \beta'_{i+1} - k_i x_0 \tan \beta'_{i+1}}{k_i - \tan \beta'_{i+1}}$$

After the above calculation, the discrete points of the intermediate aspheric curve and the side total reflection curve are obtained.

### 3.2. Model Validation

To obtain the discrete points of the central aspheric surface and the lateral reflective surface, the initial coordinate values need to be given. The critical point is in the best position to distinguish the two reflective surfaces, so the critical point is chosen as the initial value. Assuming that the maximum angle $\theta$ of the critical value is 25° and the horizontal coordinate of the initial point is 15 mm, the coordinates of each other point derived by iterating the formula are shown in Table 2.

**Table 2.** Center aspheric discrete points.

| θ | 25° | 24° | 23° | 22° | 21° | 20° | 19° | 18° | 17° | 16° | 15° | 14° | 13° |
|---|---|---|---|---|---|---|---|---|---|---|---|---|---|
| X/mm | 15.0 | 13.9993 | 13.0674 | 12.1968 | 11.3792 | 10.6089 | 9.88056 | 9.1893 | 8.5313 | 7.90279 | 7.30065 | 6.72204 | 6.16443 |
| Y/mm | 32.1676 | 31.4431 | 30.7855 | 30.1882 | 29.6440 | 29.1479 | 28.6952 | 28.2819 | 27.9046 | 27.5603 | 27.2464 | 26.9606 | 26.7010 |
| θ | 12° | 11° | 10° | 9° | 8° | 7° | 6° | 5° | 4° | 3° | 2° | 1° | 0° |
| X/mm | 5.6255 | 5.1032 | 4.5956 | 4.1011 | 3.6179 | 3.1446 | 2.6798 | 2.2222 | 1.7706 | 1.3236 | 0.8803 | 0.4395 | 0.0 |
| Y/mm | 26.46595 | 26.2538 | 26.0633 | 25.8933 | 25.7428 | 25.6110 | 25.4971 | 25.4006 | 25.3209 | 25.2576 | 25.2104 | 25.1790 | 25.0634 |

When the incident angle of the light source ($0 \le \theta \le \theta 1$), only the central aspheric reflection is available. $\theta_{max} = 25°$ and decreases in steps of 1°, and X and Y represent the horizontal and vertical coordinates of the discrete points at different $\theta$, respectively.

Similarly, when $\theta \ge 25°$, the discrete points of the lateral reflective surface can be found according to the formula, and the discrete points can be imported into Rhino software (Rhino 6, Robert McNeel & Assoc, Seattle, WA, USA) to generate a solid profile curve using coordinate point fitting, and then the solid profile curve can be rotated 360° around the main axis to obtain a 3-D model of the TIR lens as follows (Figure 8).

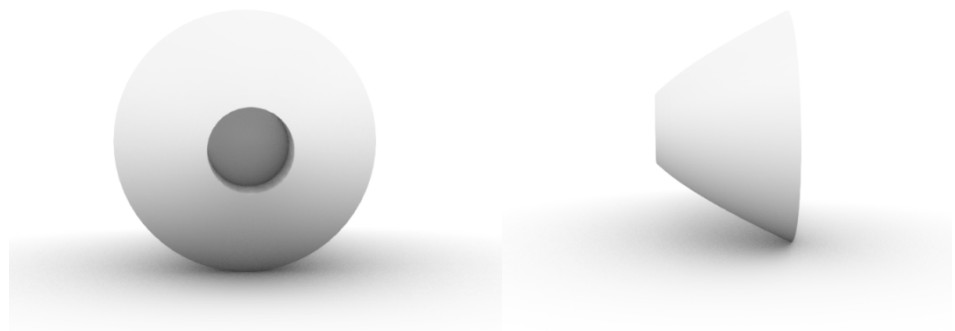

**Figure 8.** 3-D view of the TIR lens.

## 4. Simulation and Effect

### 4.1. Light Effect Design

In this paper, light tracing is performed after setting COB-packaged LEDs as point light sources. It is assumed that the light source is defined as a Lambert-type LED with a power of 16 W, and a receiving surface is set up at a distance of 2.8 m from the light source to view the illumination of the outgoing light. When there is no secondary optical design, the effect of the light source is shown in the figure below.

Figure 9 shows the light distribution curve of the LED light source, which can be intuitively seen that the initial LED lamp beads have an imitation Lambert type light distribution curve with a luminous angle of 160°. As shown in the total radiance illumination map. In the target plane range, the aperture diameter is up to 3000 mm and the distribution of illumination is not uniform, so the direct use of LED beads lighting, and the expected effect of this design cannot be achieved.

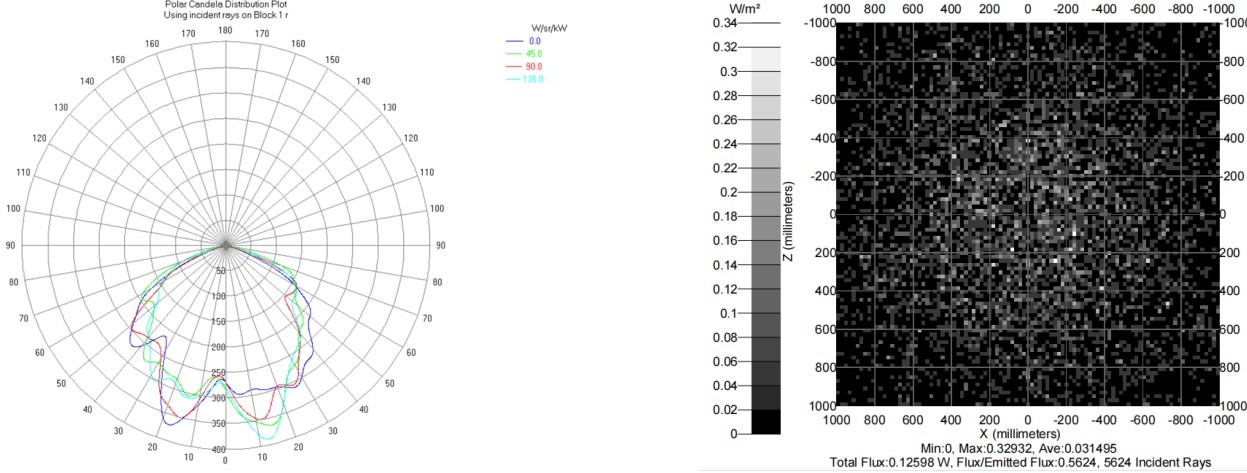

**Figure 9.** LED light distribution curve without secondary light source design.

The 3-D model was imported into Tracepro optical design software to verify the effectiveness of the optical system. The LED chip is placed at the bottom of the lens, and the ray tracing of the outgoing light from LED is performed to obtain the ray tracing diagram of the TIR lens and the illumination distribution of the outgoing surface, as shown in Figure 10.

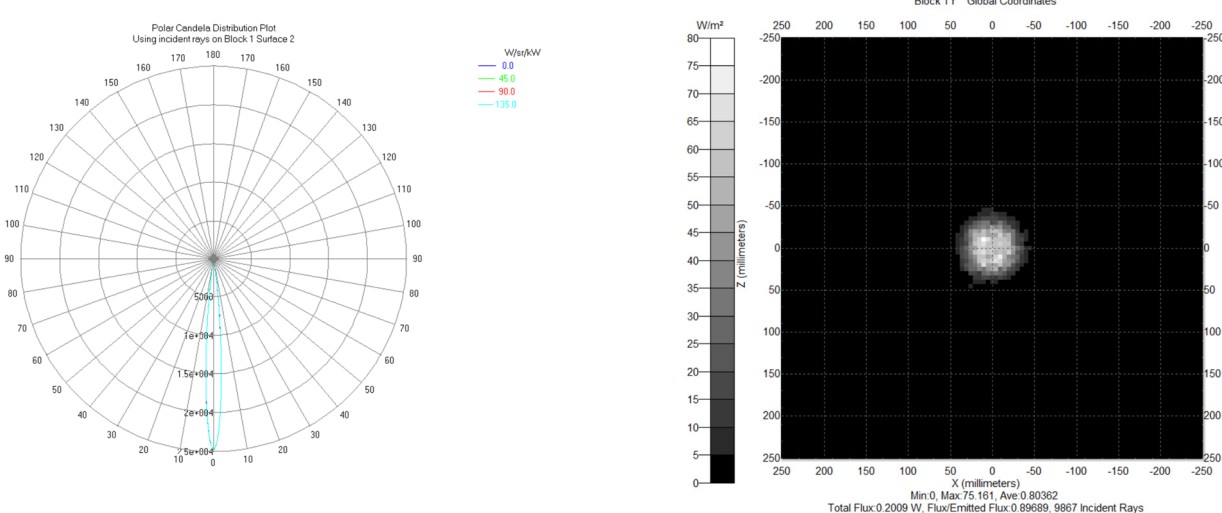

**Figure 10.** LED light distribution curve after light source design.

The TIR model obtained above was imported into the optical software Tracepro for simulation. The optical material used for the TIR lens is PMMA with a refractive index of 1.4935. After setting the LED light source as the point source, the light tracing is performed. As can be seen from Figure 10, the light energy utilization of the TIR lens is about 90%, the spot on the receiving surface is uniform, the divergence angle of the beam is small, and the diameter of the aperture is 1 m.

In order to further verify the real effect of the lawn lamp, we built the structure and application diagram of the lawn lamp in 3d Max software (3d Max 2016, Discreet, Inc., Montreal, QC, Canada). The design of LED lamps and lanterns is basically the same as the general lamp modeling procedure and method; in addition to visually reflecting the shape, size, color and material of LED lamps and lanterns, the color intensity and projection form of the light source should be simulated through application software [18]. Based on the principle of an organic combination of science and art, the structure design of the luminaire is shown below.

From left to right in Figure 11, there are the internal explosion diagram of the luminaire, the structure diagram of the luminaire, and the application diagram of the luminaire scene. In the application, it can be seen that in the light source, after the secondary optical design, the luminous beam angle of the light source becomes smaller, not only to provide the necessary good light conditions for pedestrians, but also to increase the outdoor lighting art effect.

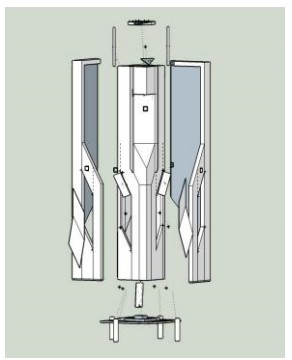 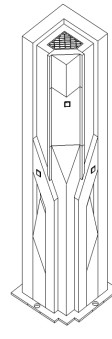 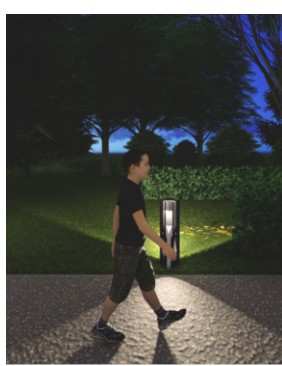

**Figure 11.** Structure design of the lawn lamp.

*4.2. Energy Saving Operations*

The outdoor lamps in this park were investigated to turn on and off the lights according to different seasons and time control [19]. Due to the large diurnal time difference between the different seasons in the region, we consider the time of day outdoor lamps are on throughout the year and obtain the data sampling period from 17:00 p.m. to 6:00 a.m. of the next day. In addition, weather factors have a great influence on the measurement results, and in order to reduce errors, the average value within a week was used as the absolute ambient light level and pedestrian flow for each time of the day. Therefore, measurements were taken from 5 September 2022 to 11 September 2022, for a duration of 1 week. The data acquisition system in this paper consists of a light sensor and a pedestrian flow sensor, which are controlled by an ultra-low-power STM32103C8T6 microcontroller. The WIFI module is used as the transmission method, and the MQTT protocol is used to transfer the collected data to the Onenet platform for data acquisition. The distribution of ambient light levels and pedestrian flow is shown in Table 3.

**Table 3.** Pedestrian flow and ambient light level data by time period.

| Time | Pedestrian Flow | Normalized Value | Environmental Illumination(lx) | Normalized Value |
|---|---|---|---|---|
| 17:00 | 159 | 0.8548 | 150 | 1 |
| 18:00 | 174 | 0.9354 | 90 | 0.5999 |
| 19:00 | 186 | 1 | 20 | 0.1333 |
| 20:00 | 90 | 0.4838 | 10 | 0.0666 |
| 21:00 | 56 | 0.3010 | 5 | 0.0333 |
| 22:00 | 21 | 0.1129 | 3 | 0.0199 |
| 23:00 | 6 | 0.0322 | 0.1 | 0 |
| 0:00 | 4 | 0.0215 | 0.04 | 0 |
| 1:00 | 2 | 0.0107 | 0.01 | 0 |
| 2:00 | 0 | 0 | 0.02 | 0 |
| 3:00 | 0 | 0 | 0.01 | 0 |
| 4:00 | 7 | 0.0376 | 0.01 | 0 |
| 5:00 | 54 | 0.2903 | 1 | 0.0133 |
| 6:00 | 112 | 0.6021 | 5 | 0.0333 |

Table 3 shows the pedestrian flow and natural light data collected in the field at ZT Park from 5:00 p.m. to 6:00 a.m. the previous day, respectively. As pedestrian traffic changes all the time, it is unreasonable to use the traditional control method, which wastes a lot of

energy. Therefore, the algorithm proposed in this study is to adjust the luminous intensity of the lawn lamp adaptively according to external factors in order to solve the problems of the prior art. Referring to Equation (1), the measured inputs are normalized to be in the range of (0, 1) and then brought into the fuzzy control to obtain the dimming coefficients for each time period, and then the power for each time period is shown as follows.

As shown in Figure 12, the blue curve represents the lighting power consumption without control, and the red curve represents the lighting power consumption when using fuzzy control. The results show that the energy saving efficiency of the control system proposed in this paper can reach 38%.

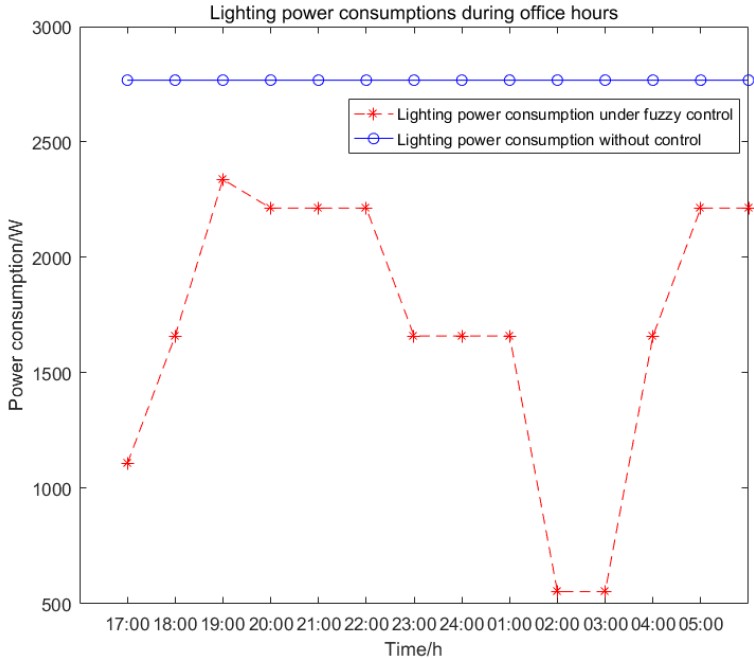

**Figure 12.** Office lighting power distribution by time period.

## 5. Conclusions

This paper takes Dalian ZT Park as the research object and addresses the current problems of the phenomenon of "long light", low light energy utilization and the substandard lighting effect of lawn lights. This paper combines artificial intelligence and optical design to propose an optimized control system for lawn lights. The system utilizes a fuzzy control algorithm with ambient light level and pedestrian flow as fuzzy inputs and dimming coefficients as fuzzy outputs to solve the unnecessary energy waste caused by the phenomenon of "long light". The proposed fuzzy control algorithm is verified by measurement data, and the results show that the system can save 38% of energy consumption. In addition, in order to improve the light energy utilization of the lawn lamp, a TIR lens was designed using the polycurved edge light principle to reduce the light source emission beam angle and enhance the lighting effect of the lawn lamp. Simulation of the designed model by Tracepro shows that the light energy utilization of the light source can be increased up to 90% compared to a light source without optical design.

It was found that the optical design of the lawn lamp light source can maximize the light energy utilization and also expand the dimming range of the lawn lamp control system to maximize energy savings. This is an important reason why this paper combines fuzzy control and optical design to optimize the lawn lamp.

**Author Contributions:** Conceptualization, X.Q.; Formal analysis, Y.F.; Investigation, M.Z.; Resources, K.L.; Funding acquisition, Z.W. All authors have read and agreed to the published version of the manuscript.

**Funding:** This research was funded by the Humanities and Social Sciences Research Project of Ministry of Education of China (Grant number: 21YJC740036) and the Dalian Academy of Social Sciences (Research Center) 2022 Annual Research Project (Grant number: 2022dlsky107).

**Institutional Review Board Statement:** Not applicable.

**Informed Consent Statement:** Not applicable.

**Data Availability Statement:** Not applicable.

**Conflicts of Interest:** The authors declare no conflict of interest.

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
