# Peer review of "Lawn Lamp Design Based on Fuzzy Control and Secondary Optical Optimization"

_applsci, doi:10.3390/app13031631_

Round 1
Reviewer 1 Report
The manuscript is well written and well organized and the results are interesting notably.
I recommend publishing it in its present form.
Author Response
Thank you very much for reviewing and giving us positive comments on the article!

Reviewer 2 Report
In the present form, the manuscript lacks coherency, as it is not clearly stated the connection between the design of the fuzzy controller and the lens. Some comments that can be helpful are stated.
1. Please define the term "lack of intelligence", described in line 76.
2. It is not clearly stated why it is necessary to introduce an adaptive system in the lawn lamps.
3. The procedure to develop the fuzzy classifier is not clearly described. For instance, do the authors use a Sugeno or mandami inference machine?
4. The experimental setup and the physical prototype description are not presented.
5. The presented results and their discussion can be improved. Does the lens mitigate the necessity of employing the designed controller? What are the advantages of employing an "intelligent lamp"? Do the electricity composition generates a saving for the users?
Author Response
Thank you very much for your revision! Please see the attachment.

Reviewer 3 Report
I have noted several minor comments to improve the work:
1) References 8 and 9 present an error in the document in the introduction section
2) More than 50% of references are outdated or more than 5 years old. Please I recommend updating them.
3) Table 3 is hard to understand, please explain it better in one paragraph and restructure or split it into two parts.
Author Response

(The authors gave the same response as above.)

Round 2
Reviewer 2 Report
The authors have addressed the reviewer's concern.